# Synchronous micromechanically resonant programmable photonic circuits

Mark Dong ®[1,2] ✉, Julia M. Boyle[1], Kevin J. Palm ®[1], Matthew Zimmermann ®[1], Alex Witte ®[1], Andrew J. Leenheer ®[3], Daniel Dominguez[3], Gerald Gilbert ®[4,7], Matt Eichenfield ®[3,5,7] & Dirk Englund ®[2,6,7]

Programmable photonic integrated circuits (PICs) are emerging as powerful tools for control of light, with applications in quantum information processing, optical range finding, and artificial intelligence. Low-power implementations of these PICs involve micromechanical structures driven capacitively or piezoelectrically but are often limited in modulation bandwidth by mechanical resonances and high operating voltages. Here we introduce a synchronous, micromechanically resonant design architecture for programmable PICs and a proof-of-principle 1×8 photonic switch using piezoelectric optical phase shifters. Our design purposefully exploits high-frequency mechanical resonances and optically broadband components for larger modulation responses on the order of the mechanical quality factor $Q_m$ while maintaining fast switching speeds. We experimentally show switching cycles of all 8 channels spaced by approximately 11 ns and operating at 4.6 dB average modulation enhancement. Future advances in micromechanical devices with high $Q_m$, which can exceed 10000, should enable an improved series of low-voltage and high-speed programmable PICs.

Programmable photonics integrated circuits[1,2] (PICs) that can generate a temporally- and spatially-varying optical signal would fulfill requirements for many optical and quantum systems. State-of-the-art PIC designs consist of meshes of reconfigurable Mach–Zehnder interferometers (MZIs), with large-scale demonstrations realized in thermo-optic[3–8], lithium niobate[9], and piezoelectric photonic platforms[10]. Several promising optical phase shifters for the construction of these mesh circuits feature microelectromechanical systems (MEMS)[11–14] with very low power consumption and scalable fabrication compatible with CMOS-foundry processes. Operation of each phase shifter often requires arbitrary electrical control across the full modulation bandwidth. This enables the execution of the entire scope (with hardware error correction[15,16]) of cascaded SU(2)[17,18] operations for either linear-optic or optical switching functions.

There are two potential issues with the conventional operation of MEMS-based PICs. The first is that, despite their scalability, micromechanical phase shifters[14,19,20] are limited in the flat modulation bandwidth (-1 μs) by the presence of mechanical eigenmodes and still typically require high voltages for either capacitive or piezoelectric actuation. The second is that for many applications, accessing the full range of unitary operations is not necessary and simply adds additional complexity. It would be sufficient for the PIC to switch periodically between a subset of particular states. For example, high-speed beam steering[21–23] would need only operate in a periodic manner to provide information on the surrounding environment for range finding applications[24–27]. Similarly, laser-scanning technologies for 3D displays[28,29] require only repeated raster scans to hold an image. In atom-based quantum systems, modular cluster-state architectures[30–32] rely on numerous attempts of remote-entanglement between select

[1]The MITRE Corporation, 202 Burlington Road, Bedford, MA 01730, USA. [2]Research Laboratory of Electronics, Massachusetts Institute of Technology, Cambridge, MA 02139, USA. [3]Sandia National Laboratories, P.O. Box 5800 Albuquerque, NM 87185, USA. [4]The MITRE Corporation, 200 Forrestal Road, Princeton, NJ 08540, USA. [5]College of Optical Sciences, University of Arizona, Tucson, AZ 85719, USA. [6]Brookhaven National Laboratory, 98 Rochester Street, Upton, NY 11973, USA. [7]These authors contributed equally: Gerald Gilbert, Matt Eichenfield, Dirk Englund. ✉e-mail: mdong@mitre.org

qubits, requiring a periodic optical switch for optical addressing of atoms as well as routing of spin-entangled photons. Periodic switches can also act as transceivers for time-multiplexed signals, with applications in processing input vector weights for optical neural networks[33] or getting around single detector latencies and enabling high-throughput photon counting[34].

If the mechanical eigenmodes inherent to MEMS-based phase shifters were purposefully designed and synchronized, we could simultaneously address both issues above. Driving each phase shifter on resonance would not only enhance the modulation strength, but naturally generate a periodic optical signal. Every modulators' eigenmodes would be engineered and synchronized to high frequency (surpassing the usual MEMS bandwidth limits) to create resonantly enhanced programmable circuits with periodic inputs and outputs. The enhancement, nominally on the order of $Q_m$, the mechanical quality factor, promises orders of magnitude decreases in each modulator's voltage-loss product, power consumption, and device footprint. The PIC electronics (needing only sinusoids and DC biases) are also simplified, no longer requiring full arbitrary control of all phase shifts. This enables narrow-band voltage sources to take advantage of electronic tank circuits for further reductions in voltage and power[35]. However, there remains an open question whether the more focused, all-periodic actuation with mechanically-resonant modulation enhancements could produce useful large-scale PICs that benefit the aforementioned applications.

Here we introduce a design architecture for optically broadband programmable PICs whose operation utilizes micromechanical resonances (Fig. 1a) for improved modulation and high-speed periodic operation in several practical configurations. We first present the theory of resonantly enhanced operation of MZI switches and direct phase modulators (Fig. 1b) for construction of phased arrays, $1 \times N$, and $N \times N$ photonic switches. We further apply the architecture by designing and fabricating a proof-of-principle resonantly enhanced $1 \times 8$ photonic switch. The switch consists of optical modulators based on piezoelectrically actuated cantilevers[14,36] (Fig. 1c), each engineered with target mechanical resonances for modulation enhancement (Fig. 1d). Lastly, we experimentally demonstrated successful operation of the PIC to show periodic

switching of all 8 channels spaced ~11 ns apart, all while leveraging an average modulation enhancement of 2.90 (4.6 dB) compared to off-resonant, low frequency actuation. Our results demonstrate the possibility of an improved series of programmable PICs operating with micromechanically resonant modulators for a broad range of applications.

## Results

### Photonic integrated circuit design architecture

The general principle of the mechanically resonant PIC architecture is to design modulators optimized for synchronized, sinusoidal drive signals at specific mechanical eigenmodes. We first briefly review the available design degrees of freedom for a typical micromechanical phase modulator (such as a cantilever-based modulator). We assume the primary phase shift $\Theta$ stems from the moving boundary[14] $x$ in which $\Theta \propto x$. In this case, we derive the modulator's response function, magnitude $A_c(\omega_s)$ and phase $\phi_c$, from the equivalent circuit of a piezoelectric oscillator[37] with a single resonance:

$$A_c(\omega_s) = (V_0/L_m)\left[(\omega_c^2 - \omega_s^2 - 2\omega_s^2\gamma\tau_0)^2 + (2\omega_s(\gamma+\gamma_s) + \omega_s\tau_0(\omega_c^2 - \omega_s^2))^2\right]^{-1/2}$$

(1)

$$\phi_c = \tan^{-1}\left(\frac{2\omega_s(\gamma+\gamma_s) + \omega_s\tau_0(\omega_c^2 - \omega_s^2)}{\omega_c^2 - \omega_s^2 - 2\omega_s^2\gamma\tau_0}\right)$$

(2)

Here, $\omega_c = \sqrt{1/L_mC_m}$ is the modulator's series resonance frequency, $\gamma = R_m/2L_m$ is the damping coefficient, $C_0$ is the device's electrical capacitance, $R_m, L_m, C_m$ are the motional circuit elements, $V_0$ is the driving amplitude, $\omega_s$ is the driving frequency, $\phi_s$ is the phase of the driving sinusoid, $R_s$ is the voltage source resistance, $\gamma_s = R_s/2L_m$ and $\tau_0 = R_sC_0$. We define the modulation enhancement factor $G_m$ imparted by the micromechanical resonance as the modulation strength on or near resonance relative to the that near DC ($\omega_s \approx 0$). Assuming the source resistance is small compared to the mechanical damping and the circuit's RC response time is faster than the

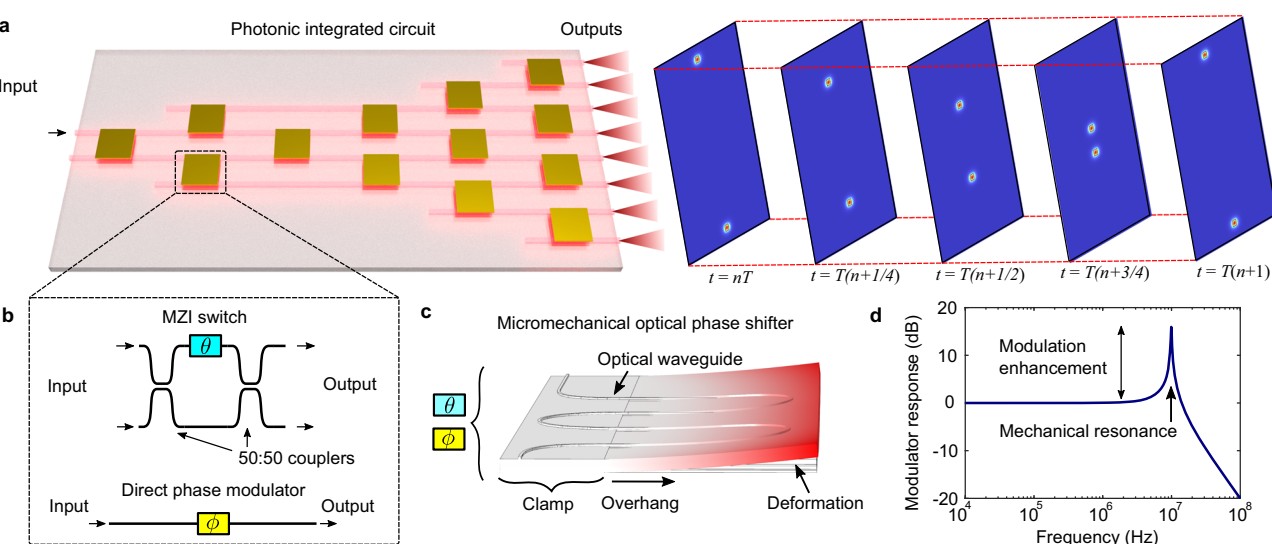

**Fig. 1 | Micromechanically resonant programmable photonic integrated circuits. a** Schematic render of a programmable photonic integrated circuit actuated with only periodic sinusoidal driving signals. The optical output channels switch periodically in time, as illustrated by the time stamps $t = nT \dots t = T(n+1)$. **b** Diagram of circuit components, consisting of Mach-Zehnder interferometer (MZI) switches and direct phase modulators with reconfigurable phases $\theta$ and $\phi$ respectively. **c** Finite-element render of a micromechanical optical phase shifter implemented as a piezo-actuated cantilever. **d** Theoretical modulator response and enhancement (Eq. 3) of a phase shifter with a single mechanical resonance where $\omega_0 = 10$ MHz and $Q_m = 40$.

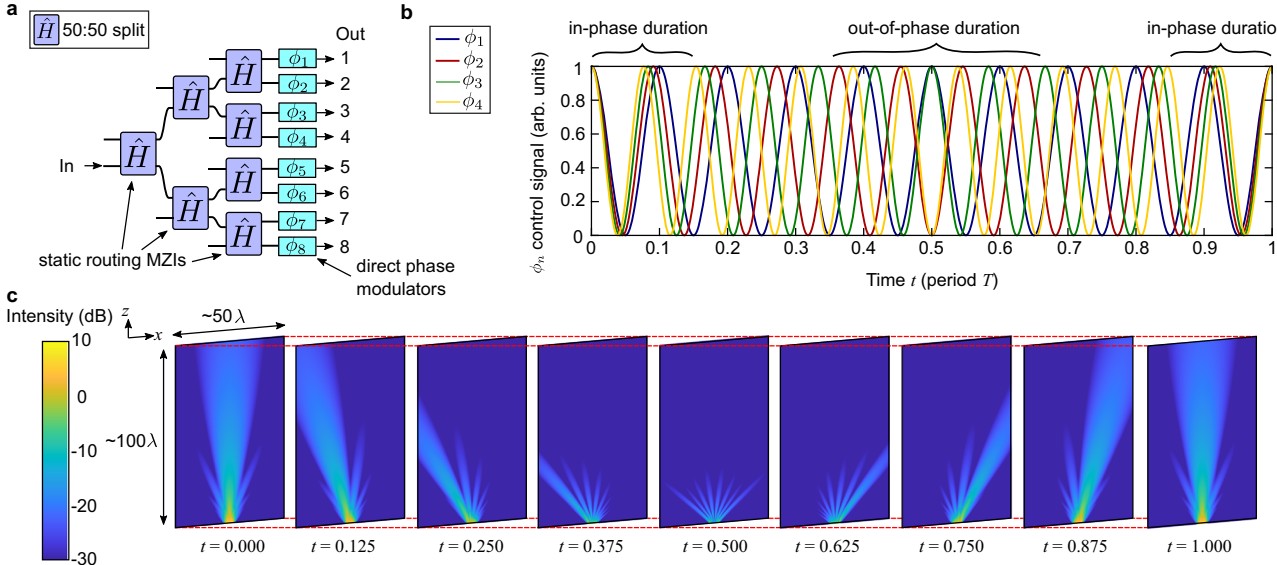

**Fig. 2 | Resonantly actuated 8-channel phased array. a** Architecture of a photonic integrated circuit (PIC) phased array based on 8 optical outputs consisting of static MZIs for power routing and direct phase modulators at the output of each channel. These modulators offset each channel's phase by a steady rate over time, defined by the frequency difference $N\Delta\omega$. **b** Example plots of the control signals applied to the first four channels $\phi_1 - \phi_4$. The channels periodically drift in and out of phase on a timescale defined by $T = 2\pi/\Delta\omega$. **c** Gaussian beam calculations of the 8 output phased array at discrete times over a spatial domain of $50\lambda \times 100\lambda$ of the $x$ and $z$ axes, respectively. The central beam combines constructively when all control phases $\phi_n$ are in phase ($t = 0$). At subsequent times, the beam sweeps counterclockwise and repeats at every $t = T$.

mechanical resonance, we calculate $G_m$ to be

$$G_m = A_c(\omega_s)/A_c(\omega_s \approx 0) = \frac{1}{\sqrt{(1 - (\omega_s/\omega_c)^2)^2 + (\omega_s/\omega_c)^2(1/Q_m)^2}} \quad (3)$$

Here, $Q_m = \frac{\omega_c}{2\gamma}$ is the mechanical quality factor. When driven on resonance such that $\omega_s \approx \omega_c$, the modulation enhancement becomes equal to the mechanical quality factor: $G_m \approx Q_m$. For the design of practical large-scale PICs, these modulator variables offer important flexibility. The cantilever parameters $\omega_c$, $Q_m$ are targeted to desired values during initial design of the modulator geometry. The operating parameters $\omega_s$, $\phi_s$, and $A_s$ are then adjusted in order to maximize the modulation enhancement $G_m$ while still synchronized with the rest of the photonic circuit. See Methods for more details of the derivation.

We proceed to illustrate how the design architecture applies to several well-known photonic circuits. A natural periodically operated device is the optical phased array, whose design and operation are shown in Fig. 2. The circuit schematic (Fig. 2a) consists of a static MZI binary tree for routing and power balancing to the $N = 8$ output channels, each with a resonantly actuated phase modulator imparting a phase $\phi_n$, defined:

$$\phi_n(t) = \pi \cos((\omega_0 + n\Delta\omega)t) \quad (4)$$

The modulation frequency is offset by $n\Delta\omega$ depending on the channel, generating a steady phase difference between channels that periodically repeats every $T = 2\pi/\Delta\omega$. Figure 2b shows example control voltages to the first four phase modulators, illustrating the phase differences plotted over time. The total modulation enhancement will be strongest when the mechanical eigenfrequency $\omega_{cn}$ match the modulation frequency of the $n$th channel.

$$\omega_{cn} = \omega_0 + n\Delta\omega \quad (5)$$

Once the phases are set, the output optical waveguides simply need to be routed, spaced, and coupled off-chip to complete the beam

scanner[38]. We numerically simulated the 1D phased array whose optical outputs are modeled as edge-emitting waveguides to show device operation. Using the phases defined in Eq. 4, the simulation calculates the Gaussian beam equation modulated in time

$$E_n(x,z,t) = E_0(1/q(z))\exp(-ik(x - x_n)^2/(2q(z)))\exp(ikz + i\phi_n(t)) \quad (6)$$

where $E_0$ is the field amplitude, $x_n$ is the $n$th output waveguide's origin point, $q(z)$ is the complex beam parameter given by $\frac{1}{q(z)} = \frac{1}{R(z)} - i\frac{\lambda}{\pi w(z)^2}$, $R(z)$ is the beam curvature whose inverse is given by $\frac{1}{R(z)} = \frac{z}{z^2 + z_R^2}$, $w(z) = w_0\sqrt{1 + z^2/z_R^2}$ is the propagation-dependent beam waist, $w_0$ is the initial beam waist at $z = 0$, $z_R = \pi w_0^2/\lambda$ is the Rayleigh range, and $\lambda$ is the wavelength. The spacing between each $x_n$ is a half wavelength and the initial beam waist is set to $w_0 = 0.5$. The total intensity is then given by

$$I(x,z,t) = \left| \sum_{n=1}^{8} E_n(x,z,t) \right|^2 \quad (7)$$

Figure 2c plots the output intensity (Eq. 7) at different times $t$, showing a periodic beam-scanner actuated with the sinusoidal driving signals corresponding to Fig. 2b. The in-phase and out-of-phase durations illustrate how the outputs interfere constructively or destructively in order to steer the beam. Compared to conventional MEMS scanners or switches[23,25,29] that typically operate with sub-MHz device speed, our proposed resonantly actuated phased array can be implemented with existing modulators[14] whose device speeds exceed >10 MHz, thus possibly enabling full field of view scan frequencies >1 MHz.

Another class of photonic circuits that benefits from periodic actuation is the integrated photonic switch. The basic $1 \times N$ switch may be implemented as a mesh of Mach-Zehnder interferometers (MZIs), as shown in Fig. 3a. Input light $E_{in}$ is coupled through a single port on the left and routed to the $n$th output $E_n$ on the right with the transfer matrix $T_{ij}(\theta_{ij})$ applied to each depth of MZIs. The normalized output

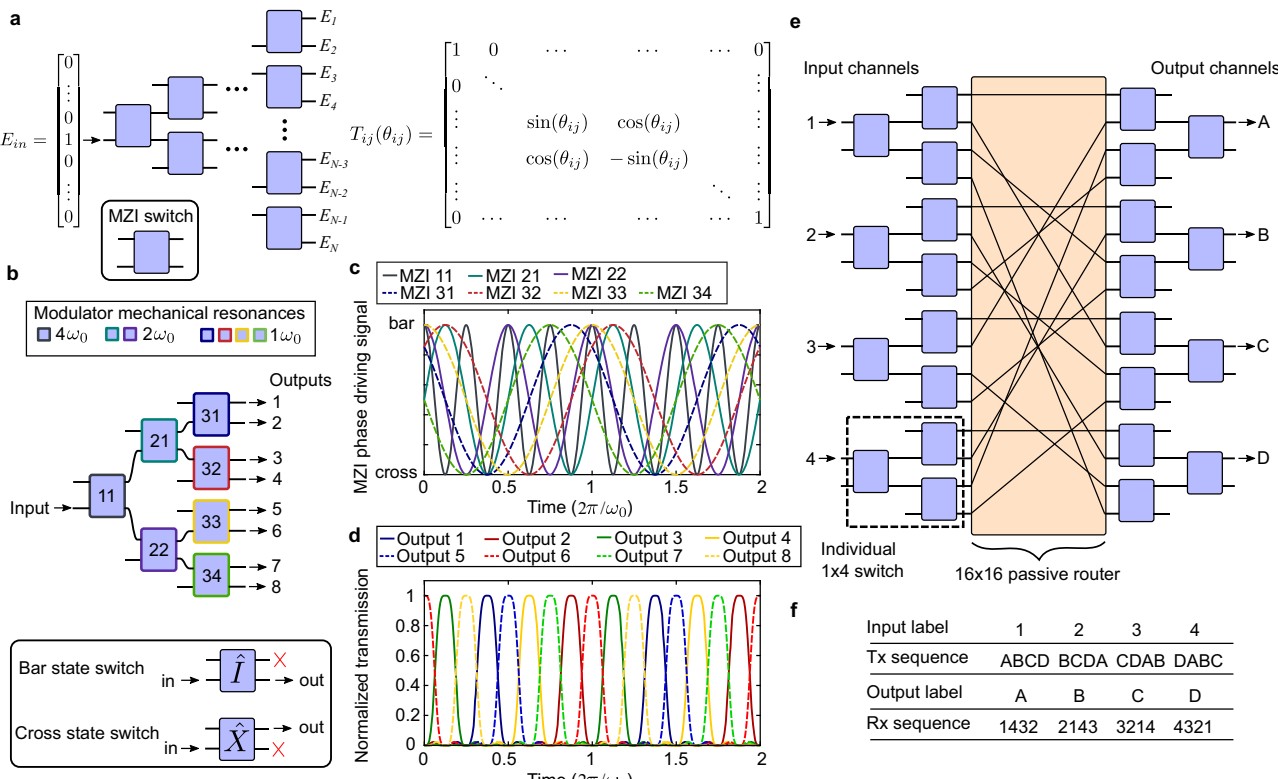

**Fig. 3 | Architecture of micromechanically resonant 1 × N and N × N integrated optical switches. a** Schematic of a general 1 × N optical switch implemented as a binary tree mesh of Mach-Zehnder interferometers (MZIs) with transfer function $T_{ij}$. **b** Design of a resonantly actuated 1 × 8 optical switch operating at three distinct frequencies, each unique to a column of MZIs. **c** Examples of driving signals in time domain to each of the MZIs with appropriate phase offsets. Each signal oscillates between bar and cross switching states. **d** Theoretical normalized optical intensity of the 8 output channels plotted in time domain when the circuit in (**b**) is driven with the signals in (**c**). **e** Schematic of a resonantly actuated 4 × 4 matrix switch consisting of 8 individual 1 × 4 switches connected through a 16 × 16 passive router. The individual switches are programmed with different driving signals to facilitate a proper switching order. **f** Example periodic transmission and receiving pattern through the 4 × 4 matrix switch.

intensities $I_n$ can be written as a product of sinusoids

$$I_n = \prod_{m=1}^{M} (1 + s(m,n) \cos(2\theta_{ij}^{(m,n)})) \qquad (8)$$

Here, $M = \log_2(N)$ the total circuit depth, $s(m,n) = \pm 1$ a sign that depends on whether the $n$th output channel's optical path is connected to the $m$th depth MZI's bar or cross port, and $\theta_{ij}^{(m,n)}$ is the time-dependent phase setting of that MZI. We illustrate a specific example of operating a resonantly actuated 1 ×8 switch (Fig. 3b). In this circuit, there are 7 MZIs labeled with the depth and the row, each with a modulator having a mechanical eigenfrequency that is an integer multiple of a base frequency $\omega_0$. All MZIs are actuated synchronously with a resonant sine wave and phase offset (Fig. 3c) such that all light is periodically routed to a single channel at specific moments in time. Figure 3d plots Eq. 8 with $N = 8$ and $M = 3$ driven by the time-dependent $\theta_{ij}^{(m,n)}$ signals in Fig. 3c, showing a periodic train of pulses emitted by each channel in sequence. The design of the optical switching alternates between the top half and the bottom half of the tree whose order can be adjusted by changing the driving phase offsets.

In addition to 1 × N switches, resonantly actuated N × N matrix switches are also possible. One representative construction of $N = 4$ is shown in Fig. 3e and generally consists of $2N$ number of 1 × N switches and a connecting N × N static optical router. The individual 1 × N switches are programmed such that in one actuation period, each input channel appears at every output channel of the N × N matrix switch. In other words, by adjusting the static connections and the sinusoid phases of the 1 × N switch, the matrix switch can be tailored to form a transmission (Tx) and receiving (Rx) sequence to suit the desired application. Figure 3f displays a table of one possible Tx and Rx sequence. The Tx sequence entries are how each input channel transmits to the corresponding output in the order of the output channels labeled. Conversely, the Rx sequence entries are the order of the input channels received by the labeled output channel. In $N = 4$ cycles, all inputs are transmitted once to all outputs.

## Micromechanically resonant 1 × 8 photonic switch

To demonstrate the micromechanically resonant architecture, we designed, fabricated, and characterized a proof-of-principle 1 × 8 photonic switch. The device is realized on a PIC platform whose layer stack consists of integrated silicon nitride waveguides and piezoelectric aluminum nitride, all fabricated on 200 mm silicon wafers[10,39]. Figure 4a illustrates the switch's resonant MZI design based on variable overhang cantilevers[14]. Each MZI consists of two independently controllable phase shifters: the first is used for DC biasing to the mid-point of the MZI response function; the second is driven sinusoidally between the approximate bar and cross states while taking advantage of any modulation enhancement from the mechanical eigenmode. We fabricated cantilevers with overhang lengths of 190.5, 95.5, and 45.5 μm, which target mechanical eigenmodes at approximately 10, 20, and 40 MHz, respectively. These cantilevers were arranged according to the schematic in Fig. 3b) with a design that corresponds to a base frequency of $\omega_0 \approx 2\pi \times 10$ MHz. After fabrication, we packaged the photonic switch on a custom PCB

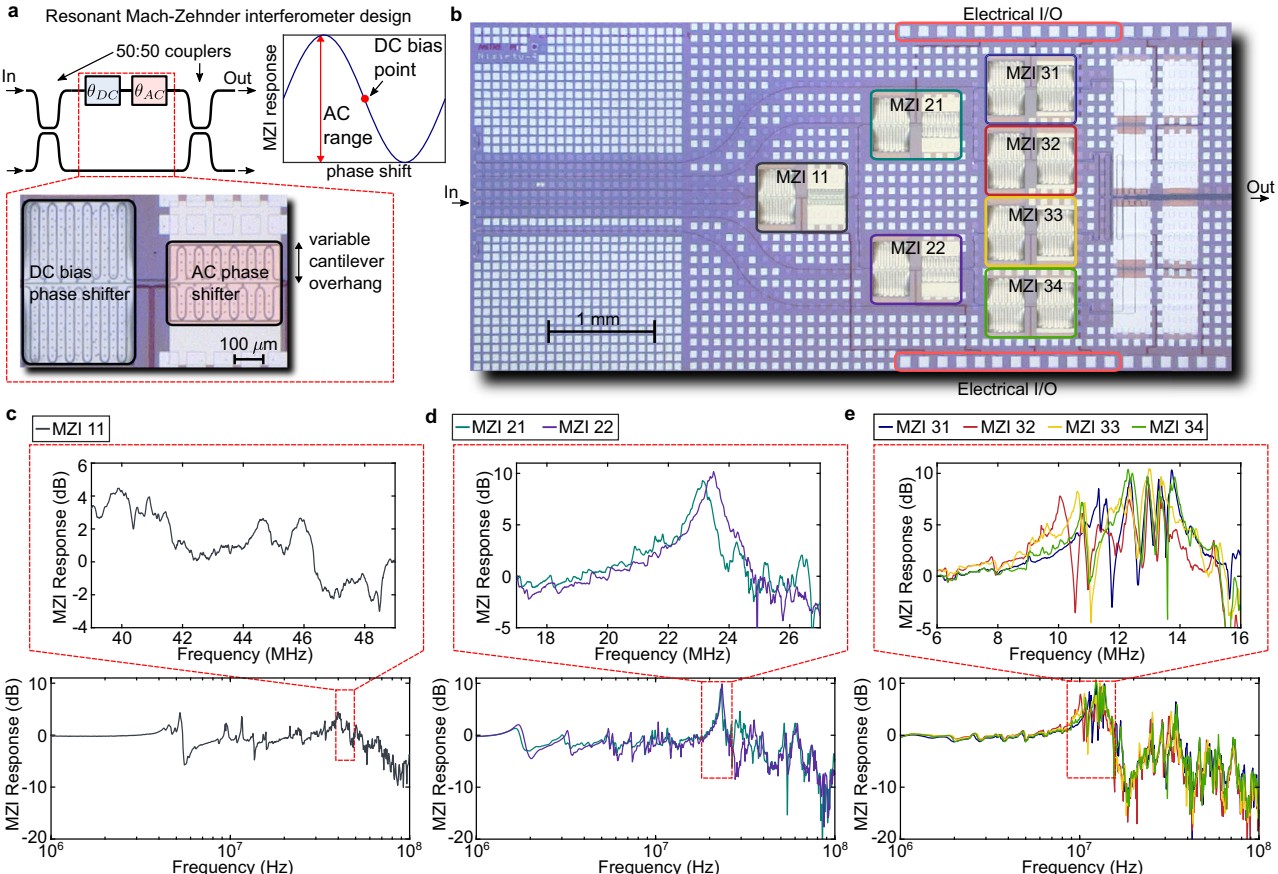

**Fig. 4 | Design and characterization of a piezoelectric 1 × 8 integrated optical switch. a** Illustration of the resonantly actuated Mach-Zehnder interferometer (MZI) design with two independent phase shifters: one larger, non-resonant cantilever for static DC biasing to the midpoint of the MZI's sinusoidal response function and one smaller, AC cantilever designed with a target mechanical eigenfrequency. **b** Optical microscope image of the fully fabricated PIC with labeled MZIs, optical inputs and outputs, and electrical bonding pads. **c–e** Measured small-signal response of the AC cantilever for the first (MZI 11), second (MZI 21 and 22) and third (MZI 31, 32, 33, and 34) columns, respectively—each measured trace is averaged 100x. In this device, the targeted base frequency is nominally 10 MHz; zoomed-in plots show the response near 40, 20, and 10 MHz, respectively.

with high-speed voltage amplifiers driven with multi-channel arbitrary waveform generators (AWGs) for electrical inputs. Optical inputs were fed through a fiber array while optical outputs were collected via a free-space imaging system for characterization. Figure 4b shows an optical microscope image of the fully processed photonic switch with labeled MZIs.

To program the circuit, we calibrated all MZIs as follows. We first measured the responses of all the DC phase shifters to store the target midpoint DC voltages. Next, we measured the small-signal, linearized response of all AC phase shifters around the DC bias point using a vector network analyzer. Figure 4c–e plots the AC actuation response of each column of MZIs averaged 100 times and normalized to the response at low frequency (100 kHz), showing clear modulation enhancement (peaks at roughly 5–10 dB) for several mechanical resonances. We note that the MZIs may still be operated as a standard binary tree switch with a DC–~1 MHz bandwidth until the non-flat frequency response would start to affect the switching fidelity. Under the micromechanically resonant architecture, however, we focus instead on the modulation response near the eigenfrequencies of interest (Fig. 4c–e insets). Despite the eigenfrequencies not precisely matching up for all MZIs due to the presence of other mechanical modes, variability in device release, and local stress gradients, the wide range of enhancement allows flexibility in choosing the base frequency $\omega_0$. Once the base frequency was chosen, we calibrated the AC $V_\pi$, the voltage required to drive the MZI between its bar and cross states with a particular modulation enhancement. Lastly, we set and synchronized

the phases $\phi_s$ of all driving signals for periodic switching operation. The PIC technology's high degree of calibration stability[10] allowed our photonic switch to operate in open loop for the duration of our experiments.

We measured the performance of the fully programmed photonic switch for a particular base frequency $\omega_0 = 11.3$ MHz, as shown in Fig. 5. This specific base frequency was selected according to the isolation of the driven mechanical eigenmode as well as the overall modulation enhancement (4.6 dB on average from Fig. 4c–e) of all 7 MZIs. Figure 5a plots time-resolved measurements using a high-speed photodiode, normalized for different losses and collection efficiencies, of all 8 output channels. The traces show clear peaks every ~11 ns which follow an 8-channel switching order that cycles every period $T = 2\pi/\omega_0$. We also plot the AC driving voltages of all MZIs (Fig. 5b) and list the values of their amplitudes $V_0$ and phase $\phi_s$ (Fig. 5c). We note that the relative phases of the driving signals differ from the ideal values in Fig. 3c as not all MZIs are driven precisely at the resonance peak—there are adjustments to $\phi_s$ to compensate for the induced, frequency-dependent phase $\phi_c$ (Eq. 2) of every individual cantilever. The switching fidelity and ordering of all channels generally matches the theoretical case as shown in Fig. 3d. However, while the maximum switching contrast during times the channels are off is approximately 0.032 +/− 0.007 (14.95 dB) for all channels, some channels show non-negligible pulsing during off-peak times (output 2 is a notable example in Fig. 5a) which reduce the switching contrast to, at worst, 8.0 dB for some channels. These off-peaks occur due to

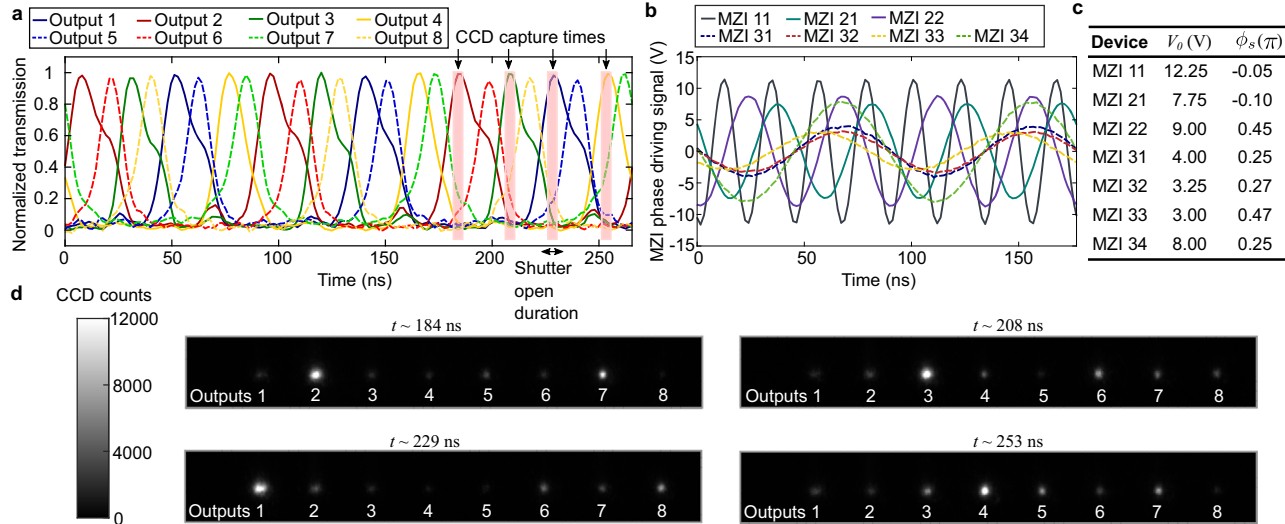

**Fig. 5 | Measurements of the 8-channel full switching sequence. a** Time-resolved measurements of all 8 optical outputs during a full switching sequence with a base frequency of 11.3 MHz for three full periods. Charged-coupled device (CCD) capture times for the imaging experiment in (**d**) are also labeled, depicting the small integration region defined by the camera shutter. **b** Plots of the sinusoidal driving inputs to the all seven Mach−Zehnder interferometers (MZIs). **c** Table of driving parameters, including voltage amplitude $V_0$ and phase $\phi_s$. The driving phase term compensates for the variation in each cantilever's phase response. **d** CCD images captured at various times $t$ as labeled in (**a**). Each image integrates the output from all 8 optical channels for 4.8 ns and illustrates an unnormalized snapshot of the output at different temporal offsets in the total switching sequence.

imperfect calibration on the larger-overhang cantilevers, which occasionally exhibit mechanical nonlinear coupling (see Supplementary Methods) over many periods between different eigenmodes.

We further characterized the switching sequence by imaging all 8 waveguide outputs simultaneously onto an intensified charge-coupled device (ICCD). The camera's image intensifier, acting as a high-speed shutter, was programmed synchronously with the MZI's driving sinusoids, allowing the capture of a small temporal window of light as illustrated in Fig. 5a. These capture times were precisely aligned to the output switching sequence by an analog gate signal generated from the same multi-channel AWGs driving the photonic switch. Figure 5d shows four CCD images at different times $t$ (relative to Fig. 3a) of a 4.8 ns exposure of the 8 output channels. Each image shows the charge count during a small subperiod (4.8 ns) of the full switching sequence, accumulated over three periods and amplified to improve the signal-to-noise ratio. The images are a measure of the unnormalized intensities of the switch's output, with bright channels matching those in the photodiode trace. Both the imaging and time-resolved experiments demonstrate the operation and viability of our 1 × 8 resonantly actuated photonic switch. See Supplementary Methods for more details of the device calibration and imaging experiment.

## Discussion

We have shown theoretically and experimentally the concept of mechanically resonant modulation for the construction of periodically operated programmable photonics. While the architecture encompasses a broad range of PIC designs and applications, there are several future improvements to the current experimental realization. The cantilever modulators' occasional nonlinear mechanical coupling is likely due to uneven undercuts and film stresses during fabrication. This can be mitigated by shortening the cantilever overhang to facilitate better undercutting and patterning the electrodes to more efficiently excite the eigenmodes of interest. Furthermore, the current photonic switch only relied on the fabricated geometry for targeting the value of the mechanical eigenmode. Thus, the overall achieved modulation enhancement fell short of the

measured maxima due to small mismatches in mechanical eigenfrequencies. The matching of mechanical eigenfrequencies across all devices on the PIC may be improved by post-process trimming and ablation[40] to adjust the mechanical spring constant or mass to optimize the cantilevers to the previously demonstrated[14] $Q_m = 40$. The overlap of output channel intensities in the full switching sequence (Fig. 5a) is reduced by adding additional frequencies to sharpen the output pulses. Furthermore, our photonics platform[10] provides a natural path for co-integration of CMOS-electronics for control of optically connected photonic chiplets in order to scale beyond current device size and yield constraints. Please see Supplementary Discussion for cantilever post-process trimming results and other circuit designs.

More broadly, the infusion of ideas from micro- and nanomechanical engineering of high-Q oscillators[41] as well as high-Q electronic oscillators should further decrease the size (offering reduced power consumption and optical losses) or operating voltage of these photonic circuits by many orders of magnitude. Devices in other thin-film piezoelectric MEMS platforms have shown mechanical quality factors exceeding 10,000[42,43] or even higher quality factors of $10^6$–$10^9$ with engineered structures[44,45] and tuning techniques[46], operating at kHz−MHz resonance frequencies. Moreover, geometries can also be optimized for speed, with demonstrations reaching GHz eigenfrequencies[47,48] with still high mechanical quality factors greater than 100. Future advances to resonant MEMS with integrated photonics, an improved category of low-power and high-speed programmable PICs should be possible.

## Methods

### Linear response function of a piezoelectric optical modulator

We model the response function of our piezoelectrically actuated optical modulators using the standard equivalent circuit of a piezoelectric device[37] (Butterworth−van Dyke model). The circuit has a frequency-dependent effective impedance $Z_{eq}(\omega)$ given by

$$Z_{eq}(\omega) = R_s + \frac{\omega_c^2 - \omega^2 + 2j\omega\gamma}{j\omega/L_m + j\omega C_0(\omega_c^2 - \omega^2 + 2j\omega\gamma)} \quad (9)$$

where $j$ is the imaginary unit, $R_s$ is the series source resistance, $\omega_c = \sqrt{1/L_m C_m}$ is the series resonance frequency, $\gamma = R_m/2L_m$ is the damping coefficient, $C_0$ is the device's electrical capacitance, and $R_m$, $L_m$, $C_m$ are the motional circuit elements. The frequency-dependent applied voltage $V_s(\omega)$ and current $i(\omega)$ are defined

$$V_s(\omega) = V_0 \pi \left[ e^{-j\phi_s} \delta(\omega - \omega_s) + e^{j\phi_s} \delta(\omega + \omega_s) \right] \tag{10}$$

$$i(\omega) = V_s(\omega)/Z_{eq}(\omega) \tag{11}$$

We are interested in the charge accumulated on the motional capacitor $q_m$, which is analogous to the motional displacement $x$ responsible for the optical phase shift. We make the assumption that the phase shift $\theta$ ultimately imparted by the modulator linearly follows the displacement, i.e., $x = B_1 q_m$ and $\theta = B_2 q_m$ where $B_1, B_2$ are constants representing the optomechanical coupling. We calculate the current flowing through the motional circuit elements:

$$i_m(\omega) = i(\omega) \frac{1/j\omega C_0}{1/j\omega C_0 + j\omega L_m + R_m + 1/j\omega C_m} \tag{12}$$

and using the relation $i_m(\omega) = j\omega q_m(\omega)$, we find

$$q_m(\omega) = \frac{V_s(\omega)}{L_m} \left[ \frac{1}{j\omega R_s/L_m + j\omega R_s C_0(\omega_c^2 - \omega^2 + 2j\omega\gamma) + \omega_c^2 - \omega^2 + 2j\omega\gamma} \right] \tag{13}$$

To find the time-dependent motional displacement, we first introduce $\gamma_s = R_s/2L_m$ and $\tau_0 = R_s C_0$ for simplicity. Inserting Eq. 10 into Eq. 13 and performing the inverse Fourier transform, we calculate the time-dependent charge to be

$$q_m(t) = \frac{1}{2\pi} \int_{-\infty}^{\infty} d\omega e^{-j\omega t} q_m(\omega) = A(\omega_s) \cos(\omega_s t + \phi_s + \phi_c(\omega_s)) \tag{14}$$

$$A_c(\omega_s) = (V_0/L_m) \left[ (\omega_c^2 - \omega_s^2 - 2\omega_s^2 \gamma \tau_0)^2 + (2\omega_s(\gamma + \gamma_s) + \omega_s \tau_0(\omega_c^2 - \omega_s^2))^2 \right]^{-1/2} \tag{15}$$

$$\phi_c = \tan^{-1} \left( \frac{2\omega_s(\gamma + \gamma_s) + \omega_s \tau_0(\omega_c^2 - \omega_s^2)}{\omega_c^2 - \omega_s^2 - 2\omega_s^2 \gamma \tau_0} \right) \tag{16}$$

We define the modulation enhancement factor $G_m$ by normalizing the actuation amplitude $A_c(\omega_s)$ to that near DC ($A_c(\omega_s \approx 0)$).

$$A_c(\omega_s \approx 0) = \frac{V_0}{\omega_c^2 L_m} \tag{17}$$

$$G_m = A_c(\omega_s)/A_c(\omega_s \approx 0)$$
$$= \left[ \left( 1 - \left(\frac{\omega_s}{\omega_c}\right)^2 - \left(\frac{\omega_s}{\omega_c}\right)^2 \frac{\omega_c \tau_0}{Q_m} \right)^2 + \left( \frac{\omega_s}{\omega_c Q_m}(1 + R_s/R_c) + \omega_s \tau_0 \left( 1 - \left(\frac{\omega_s}{\omega_c}\right)^2 \right) \right)^2 \right]^{-1/2} \tag{18}$$

where we have inserted $Q_m = \omega_c/2\gamma$ the mechanical quality factor. If we assume operating conditions of (i) small source resistance compared to the mechanical damping ($R_s \ll R_m$) and (ii) the mechanical resonance is far below the circuit's RC response ($\tau_0 \omega_c, \tau_0 \omega_s \ll 1$), then the modulation enhancement simplifies considerably to

$$G_m = \frac{1}{\sqrt{(1 - (\omega_s/\omega_c)^2)^2 + (\omega_s/\omega_c)^2(1/Q_m)^2}} \tag{19}$$

which is Eq. 3 in the main text.

## Data availability

The data that support the plots within this paper are available under restricted access due to MITRE's information security policies. Access can be obtained from the corresponding authors upon request.

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

## Acknowledgements

Major funding for this work is provided by MITRE for the Quantum Moonshot Program. D.E. acknowledges partial support from Brookhaven National Laboratory, which is supported by the U.S. Department of Energy, Office of Basic Energy Sciences, under Contract No. DE-SC0012704 and the NSF RAISE TAQS program. M.E. performed this work, in part, with funding from the Center for Integrated Nanotechnologies, an Office of Science User Facility operated for the U.S. Department of Energy Office of Science. M.D. and M.Z. thank L. Chan, K. Dauphinais, and S. Vergados for their support in constructing and testing the mechanical and electronic components. M.D. thanks T. Poimenidis for additional technical insights regarding the PIC control electronics.

## Author contributions

M.D., J.M.B., K.J.P., M.Z. and A.W. built the experimental setups and performed the experiments. M.D. performed the theoretical analysis and designed the photonic integrated circuit. M.E. and A.J.L., with assistance from D.D., supervised the device fabrication. M.Z. and M.D. designed and programmed the electronic control system. M.D., M.E., and D.E. conceived the programmable PIC architecture. G.G., M.E. and D.E. supervised the project. M.D. wrote the manuscript with input from all authors.

## Competing interests

D.E. is a Scientific Advisor to and holds shares in QuEra Computing. The other authors declare no competing interests.
