## [Peer Review File · Nature Communications]

Synchronous micromechanically resonant programmable photonic circuitsEditorial Note: This manuscript has been previously reviewed at another journal that is not operating a transparent peer review scheme. This document only contains reviewer comments and rebuttal letters for versions considered at *Nature Communications*.

REVIEWERS' COMMENTS

Reviewer #1 (Remarks to the Author):

The authors provided detailed and satisfactory clarifications to several issues I previously pointed out in the revised manuscript. Although I am still not convinced that the reported approach has clear advantages over the conventional ones, it is nevertheless an approach worth further exploration by the research community in hopes that the associated technical challenges can be resolved or mitigated effectively. Therefore, I recommend the acceptance of this manuscript for publication in *Nature Communications*.

Reviewer #3 (Remarks to the Author):

I am glad to be informed that the authors have addressed all the reviewers' comments and submitted the revised version to *Nature Communications*. In this round, I consider the acceptance of the manuscript transfer made by the Editor. I think that the authors made a very good job in revising the manuscript. As for the current version, I admit that the qualities of organization and technical writing become excellent. The revisions made in text and supplementary information are overall good. Addition of Supplementary Fig. 7 and detail of the post process trimming technologies reinforced originality of the manuscript. In conclusion, I recommend considering publication in *Nature Communications* after the authors would consider the following minor comments.

- (1) Despite that I have pointed in comment (9) in the previous round, they still describe in Abstract that by further leveraging micromechanical devices with high Q_m , which can exceed 1 million. In the revised version, the updated mechanical Q in their platform is still around 10,000. They should revise 1 million as 10,000, otherwise they should demonstrate that mechanical Q of 1 million can be theoretically possible with keeping phase shift amplitude and high modulation speed (\sim GHz). Basically, the word "leveraging" makes the text ambiguous.
- (2) Although this does not affect the text, I think that their answer to comment (3) of Reviewer #2 is not appropriate. Why do they need a 1×512 channel switch? Even the device has bandwidth of GHz, it is difficult for the proposed resonant 1×512 photonics switch to achieve 10 MHz operation speed (Base frequency ω_0 is inverse of 2^8 of the bandwidth. The high-frequency MZI 00 presented in Supplementary Fig. 6 needs $512 \times \omega_0$.) Note that in Fig. 3(e), 4×4 matrix switch does not consist of 1×16 (1×8) switches but many 1×4 switches. I think that scalability of 1×512 and the high mechanical Q of 1 million are not minimum requirements of publication in *Nature Communication*. So, I recommend them to claim reasonable theoretical values.
- (3) In page 5, after the line 146, I recommend the authors to add sentences to describe the merit of using their platform for phased array against other conventional MEMS platforms.

Please see our responses to the final reviewer comments below. Original reviewer comments are in italics while our own responses are in regular text. Changes in our revised manuscript are in red text.

Reviewer #1 (Remarks to the Author):

The authors provided detailed and satisfactory clarifications to several issues I previously pointed out in the revised manuscript. Although I am still not convinced that the reported approach has clear advantages over the conventional ones, it is nevertheless an approach worth further exploration by the research community in hopes that the associated technical challenges can be resolved or mitigated effectively. Therefore, I recommend the acceptance of this manuscript for publication in Nature Communications.

We thank the reviewer once more and we appreciate the time taken to evaluate our work.

Reviewer #3 (Remarks to the Author):

I am glad to be informed that the authors have addressed all the reviewers' comments and submitted the revised version to Nature Communications. In this round, I consider the acceptance of the manuscript transfer made by the Editor. I think that the authors made a very good job in revising the manuscript. As for the current version, I admit that the qualities of organization and technical writing become excellent. The revisions made in text and supplementary information are overall good. Addition of Supplementary Fig. 7 and detail of the post process trimming technologies reinforced originality of the manuscript. In conclusion, I recommend considering publication in Nature Communications after the authors would consider the following minor comments.

We thank the reviewer once more and we appreciate the time taken to evaluate our work.

(1) Despite that I have pointed in comment (9) in the previous round, they still describe in Abstract that by further leveraging micromechanical devices with high Q_m , which can exceed 1 million. In the revised version, the updated mechanical Q in their platform is still around 10,000. They should revise 1 million as 10,000, otherwise they should demonstrate that mechanical Q of 1 million can be theoretically possible with keeping phase shift amplitude and high modulation speed (~GHz). Basically, the word "leveraging" makes the text ambiguous.

Based on this feedback and Nature Communications formatting guidelines, we have rewritten the abstract (shortening to 150 words) and removed the word “leveraging” to make it less ambiguous and revised the mechanical Q to 10000. We also removed the word “leveraging” in the concluding paragraph of the paper.

(2) Although this does not affect the text, I think that their answer to comment (3) of Reviewer #2 is not appropriate. Why do they need a 1x 512 channel switch? Even the device has bandwidth of GHz, it is difficult for the proposed resonant 1 x 512 photonics switch to achieve 10 MHz operation speed (Base frequency ω_0 is inverse of 2^8 of the bandwidth. The high-frequency MZI 00 presented in Supplementary Fig. 6 needs 512 x ω_0 .) Note that in Fig. 3(e), 4 x 4 matrix switch does not consist of 1 x 16 (1x 8) switches but many 1 x 4 switches. I think that scalability of 1 x 512 and the high mechanical Q of 1 million are not minimum requirements of publication in Nature Communication. So, I recommend them to claim reasonable theoretical values.

We do agree with the reviewer that the challenges with resonance engineering would be, by far, the near-term problem to solve long before we reach 512 channels. Our answer was meant for illustrative purposes that device size (which is easily extrapolated) is not the current limiting factor. We will be more modest with our estimates in the future.

(3) In page 5, after the line 146, I recommend the authors to add sentences to describe the merit of using their platform for phased array against other conventional MEMS platforms.

We added the following sentence to compare our platform with some MEMS-based phased arrays:

“Compared to conventional MEMS scanners or switches^{23,25,29} which typically operate with sub-MHz device speed, our proposed resonantly actuated phased array can be implemented with existing modulators¹⁴ whose device speeds exceed >10 MHz, thus possibly enabling full field of view scan frequencies >1 MHz.”